# Real-World Safety and Effectiveness of Voretigene Neparvovec: Results up to 2 Years from the Prospective, Registry-Based PERCEIVE Study

**DOI:** 10.3390/biom14010122

**Published:** 2024-01-17

**Authors:** M. Dominik Fischer, Francesca Simonelli, Jayashree Sahni, Frank G. Holz, Rainer Maier, Christina Fasser, Andrea Suhner, Daniel P. Stiehl, Bee Chen, Isabelle Audo, Bart P. Leroy

**Affiliations:** 1Centre for Ophthalmology, University of Tübingen, 72076 Tübingen, Germany; 2Oxford Eye Hospital, Oxford University NHS Foundation Trust, Oxford OX3 9DU, UK; 3Nuffield Laboratory of Ophthalmology, Nuffield Department of Clinical Neurosciences, University of Oxford, Oxford OX3 9DU, UK; 4Eye Clinic, Multidisciplinary Department of Medical, Surgical and Dental Sciences, University of Campania Luigi Vanvitelli, Via S. Pansini, 5, 80131 Napoli, Italy; francesca.simonelli@unicampania.it; 5Novartis Pharma AG, 4056 Basel, Switzerland; jayashree.sahni@novartis.com (J.S.); asuhner@yahoo.com (A.S.); dpstiehl@gmail.com (D.P.S.); 6Department of Ophthalmology, University of Bonn, Ernst-Abbe-Straße 2, 53127 Bonn, Germany; frank.holz@ukbonn.de; 7Retina International, D02 TW98 Dublin, Ireland; Retina Suisse, 8005 Zürich, Switzerland; cfasser@e-link.ch; 8Novartis Pharmaceutical Corporation, East Hanover, NJ 7936, USA; blchen999@gmail.com; 9Sorbonne Université, INSERM, CNRS, Institut de la Vision, 75012 Paris, France; isabelle.audo@inserm.fr; 1015–20 Hôpital National de la Vision, National Rare Disease Center REFERET, INSERM-DGOS CIC1423, 75012 Paris, France; 11Department of Ophthalmology & Center for Medical Genetics Ghent, Ghent University & Ghent University Hospital, 9000 Ghent, Belgium; bart.leroy@ugent.be; 12Children’s Hospital of Philadelphia, Philadelphia, PA 19104, USA

**Keywords:** effectiveness, real world, *RPE65*-associated inherited retinal dystrophies, safety, voretigene neparvovec

## Abstract

Voretigene neparvovec (VN) is the first available gene therapy for patients with biallelic *RPE65*-mediated inherited retinal dystrophy who have sufficient viable retinal cells. PERCEIVE is an ongoing, post-authorization, prospective, multicenter, registry-based observational study and is the largest study assessing the real-world, long-term safety and effectiveness of VN. Here, we present the outcomes of 103 patients treated with VN according to local prescribing information. The mean (SD) age was 19.5 (10.85) years, 52 (50.5%) were female, and the mean (SD) duration of the follow up was 0.8 (0.64) years (maximum: 2.3 years). Thirty-five patients (34%) experienced ocular treatment-emergent adverse events (TEAEs), most frequently related to chorioretinal atrophy (*n* = 13 [12.6%]). Eighteen patients (17.5%; 24 eyes [13.1%]) experienced ocular TEAEs of special interest, including intraocular inflammation and/or infection related to the procedure (*n* = 7). The mean (SD) changes from baseline in full-field light-sensitivity threshold testing (white light) at month 1, month 6, year 1, and year 2 were −16.59 (13.48) dB (51 eyes), −18.24 (14.62) dB (42 eyes), −15.84 (14.10) dB (10 eyes), and −13.67 (22.62) dB (13 eyes), respectively. The change in visual acuity from baseline was not clinically significant. Overall, the outcomes of the PERCEIVE study are consistent with the findings of VN pivotal clinical trials.

## 1. Introduction

Inherited retinal dystrophies (IRDs) are rare heterogeneous disorders characterized by progressive retinal degeneration and vision loss [1]. Mutations that can lead to IRDs have been identified in >300 genes [2]. Biallelic mutations in the *RPE65* gene (online mendelian inheritance in man (OMIM) * 180069) cause the most severe forms of IRD, namely, Leber congenital amaurosis (LCA) and early onset retinal dystrophy, with an estimated worldwide prevalence of 1.20–2.37 per 100,000 [3]. The *RPE65* gene is selectively expressed in retinal pigment epithelium (RPE) cells and encodes the 65-kDa retinoid isomerohydrolase, which converts all-*trans*-retinyl ester to 11-*cis*-retinal, an essential step in the regeneration of visual pigment [4,5]. Clinical manifestations of *RPE65* mutation-associated IRDs include night blindness, a concentrically constricted visual field (VF), and reduced best-corrected visual acuity (BCVA) [6]. The progressive, irreversible loss of RPE and retinal photoreceptor cells leads to severe visual dysfunction and eventually results in complete blindness [7]. The most common diagnoses of biallelic *RPE65* mutation-associated IRDs are LCA (LCA2, OMIM # 204100) and retinitis pigmentosa (RP20, OMIM # 613794) [5,8].

Voretigene neparvovec (VN; AAV2-hRPE65v2) is the first approved ocular gene augmentation therapy in patients with visual impairments, due to biallelic *RPE65* mutation-associated IRDs, who have sufficient viable retinal cells [9,10]. VN was approved by the US Food and Drug Administration (FDA) in 2017 and later by the European Medicines Agency (EMA) in 2018 [9,10,11,12]. It is a recombinant adeno-associated virus vector serotype 2 (AAV2) containing human *RPE65* cDNA with an altered Kozak sequence and a chicken β-actin promoter with a cytomegalovirus enhancer [13]. VN delivers *RPE65* cDNA in RPE cells and induces the production of a functional RPE65 enzyme, thus allowing the restoration of the visual cycle [4].

The current evidence on the safety and efficacy of VN is from two open-label, phase 1 trials (NCT00516477, NCT01208389) and an open-label, randomized-controlled, phase 3 trial (NCT00999609), in which, a total of 41 patients received VN [14]. In these trials, VN administration to patients with biallelic *RPE65*-associated IRD led to improvements in functional vision (multi-luminance mobility test) and visual function (full-field light-sensitivity threshold (FST) and VF) [13,14,15]. Furthermore, the results show an acceptable safety profile, which is consistent with vitrectomy and the subretinal administration procedure [11,13,14].

However, the real-world data from the post-authorization period on the safety and effectiveness of VN are limited [16,17,18,19,20]. Post-authorization safety studies (PASSs) are necessary to obtain further information on a medicine’s safety, or to measure the effectiveness of risk-management measures, and the PERCEIVE study was designed to address this need.

The observational nature of this PASS study (i.e., a clinical trial protocol focusing on safety, without standard operating procedures for data collection across sites, with no pre-defined or mandated visit schedule, and the lack of a central reading center) will inevitably result in some differences in the data collection across study sites. Despite this limitation, the current analysis of PERCEIVE study data provides the first global (ex-US) snapshot of the VN safety profile within a real-world setting.

This study is, to our knowledge, the largest complete cohort collection of safety and outcome information on the first approved ocular gene therapy in patients with biallelic *RPE65* mutation-associated IRDs in a clinical setting. The datum from the PERCEIVE study is an essential information resource of the scientific evidence for ocular gene therapy with VN.

## 2. Methods

### 2.1. Study Design

PERCEIVE is a prospective, longitudinal, multicenter, multinational (ex-US), post-authorization safety study (PASS; European Union electronic Register of Post-Authorization Studies (https://www.encepp.eu/encepp/viewResource.htm?id=37005: accessed on 5 January 2024). The study commenced in December 2019 with an enrollment period of 5 years and is planned for completion in December 2029. Each patient enrolled will be followed for a 5-year period after VN administration. The study sites are VN treatment centers with qualified vitreoretinal surgeons and pharmacists (who have participated in the mandatory educational program on the use of VN), and IRD referral centers.

### 2.2. Study Population and Treatment

Participation in the study was voluntary; whenever possible, the patients were enrolled prior to treatment with VN. Patients who were either scheduled to receive VN or had already been treated in at least 1 eye, and who had neither previously participated nor are currently participating in an interventional clinical trial with VN were included in the study. Patients could withdraw from the study at any time without prejudice to their care. VN was administered according to the local prescribing information, and patients were followed up as per routine medical practice.

### 2.3. Study Objectives and End Points

The primary objective of PERCEIVE is to characterize the long-term safety profile of VN via the systematic collection of protocol-defined treatment-emergent adverse events (TEAEs) of special interest (TEAESI), and any other TEAEs or serious TEAEs. TEAESIs included ocular TEAEs, a lack of efficacy and/or decline of efficacy over time, third-party transmission, host immune response, and the development or exacerbation of oncologic, hematologic, neurologic, or autoimmune diseases.

The secondary objectives were (1) to observe pregnancy outcomes in patients (and female partners of patients) who received VN; and (2) to assess visual function over time, as measured by VA (visual acuity), VF, FST for white light, and optical coherence tomography (OCT) examinations.

### 2.4. Data Collection

After the patients received VN in at least 1 eye, follow-up visits were scheduled as part of the routine medical care. The following information was collected: demographic data, medical and surgical history, diagnosis, VN administration details, TEAESI/TEAE/serious TEAEs, ophthalmic examinations (VA, VF, FST, and OCT, measured according to the method used by the site as per routine care), pregnancy outcomes (in a patient or female partners of a patient), and concomitant medications and procedures. Although the data collection was performed using case report forms, data reporting varied among the study sites as there were no standardized or mandated follow ups, standardized protocols for the data collection, or central image reading centers for the assessments.

VA was assessed as per standard of care and converted to the logarithm of the minimum angle of resolution (LogMAR) scores for the analysis. Off-chart vision measurements were assigned values using the scale adapted from Lange 2009 [21]. VF results from Octopus or Goldmann perimetry were collected for kinetic fields, as available, and from Humphrey and Octopus computerized testing for macular static fields, including foveal sensitivity thresholds. In addition, the study collected microperimetry results, if available. If OCT was performed, the data from central macular and/or peripheral scans were collected, including retinal/foveal thickness, macular volume, subretinal and intraretinal fluids, information on the outer nuclear layer thickness (presence and thickness), and the ellipsoid zone (intact/disrupted/absent). OCT data were provided by the centers based on the data collected from the respective case report forms. The data or quality of scans were not evaluated by a central reading center. In this analysis, all OCT data were acquired using Heidelberg Spectralis OCT. The data sources used included medical notes, electronic medical records, hospital discharge files documented during routine care, and ophthalmic assessments (VA, VF, FST, and OCT). A standardized safety questionnaire was used to collect TEAESI information. All the data were collected at a minimum on an annual basis after treatment.

### 2.5. Statistical Analysis

The full analysis set (FAS), which included all enrolled individuals who received VN in at least 1 eye and provided informed consent, was used to summarize all the data. Continuous variables were summarized in terms of the mean, SD, median, and minimum and maximum values, and categorical variables were summarized using frequency counts and percentages. Non-ocular and ocular TEAEs were summarized using the number and percentage of patients or affected eyes, and the number of events by primary system organ class (SOC) and preferred term (PT). If a patient/treated eye reported the same TEAE PT/SOC more than once, only the first occurrence of that PT/SOC for the patient/treated eye was counted. For the VA, VF, FST, and OCT examinations, the observed values at month 1, month 6, year 1, and year 2, and the change in baseline (BL) for the same timepoints were summarized using descriptive statistics for the 1st-treated eye and the 2nd-treated eye separately.

## 3. Results

A total of 106 patients were enrolled in the study until August 2021 (data cut-off). Of them, 103 patients (183 eyes) received VN and were included in the FAS (Appendix A). Until the year 1 analysis (data cut-off: August 2020), 15 patients were enrolled, and 91 patients were enrolled between August 2020 and 2021. The majority of the patients (*n* = 60; 58.3%) were enrolled prior to treatment with VN, whereas 19 patients (18.4%) were enrolled after treatment in 1 eye and 24 patients (23.3%) were enrolled after treatment in both eyes. The mean (SD) duration of the follow up was 0.8 (0.64) years (maximum: 2.3 years). The majority of the patients had less than a year of follow-up session (*n* = 79; 76.7%); 24 patients (23.3%) had >1 year of follow up and 7 patients (6.8%) had >2 years of follow up. However, the data for all the study variables were not available at all timepoints for the analysis owing to the inconsistent reporting of data.

### 3.1. Demographics and Baseline Characteristics

The mean (SD) age of patients was 19.5 (10.85) years at the time of the VN administration (age < 18 years: *n* = 51 [49.5%]; age ≥ 18 years: *n* = 52 [50.5%]; age range = 2–51 years). Fifty-two patients (50.5%) were females (Table 1). Patients were enrolled from 15 countries, mostly from Germany (23.3%), France (20.4%), and Italy (14.6%; Table 1). At BL, the data on central subfield thickness using a spectral domain OCT scan were reported for 135 eyes. The presence of an outer nuclear layer was reported in the central subfield scan for 109 eyes (80.7%). Ellipsoid zone data from the central subfield scans were available for 128 eyes: the ellipsoid zone was intact in 41 (32%), disrupted in 69 (53.9%), and absent from 18 (14.1%) treated eyes. Among patients aged <18 years (66 eyes), the ellipsoid zone line was intact in 27 (40.9%) and disrupted in 39 (59.1%) of the treated eyes; an absence of the ellipsoid zone line was not reported in this age group.

### 3.2. Treatment Intervention

Overall, 103 patients (183 eyes) were treated with VN until the time of the data cut-off. Eighty patients (77.7%) received bilateral treatment and 23 patients (22.3%) received unilateral treatment. The majority of the patients (158 eyes; 86.3%) received the recommended dose of 300 µL of VN; 22 eyes (12%) received <300 µL of VN, whereas 1 eye received >300 µL. The reason reported for administering lower doses of VN was usually safety concerns, such as avoiding macular hole formation in the presence of atrophic areas. In most of the cases, a single retinotomy site (132 eyes; 72.1%) and a single bleb (134 eyes; 73.2%) were created. Deviations from the standard procedure (per prescribing information) presented by the sites included the use of an automated injection system (43 eyes; 23.5%). Other deviations, 26 eyes (14.2%) included the internal limiting membrane peeling at the injection site (19 eyes; 10.4%), use of triamcinolone to identify an incomplete vitreous separation (3 eyes; 1.6%), and use of trypan blue for limiting dissections at the injection site (3 eyes; 1.6%). The median time between the first and second eye treatments was 7 (range: 7–476) days.

### 3.3. Effectiveness

The mean (SD) FST was −4.56 (10.88) dB (127 eyes) at BL. An increase in white-light sensitivity from BL was reported as early as month 1’s (mean [SD] change in BL: −16.59 [13.48] dB; 51 eyes); the highest value was reported in month 6’s (mean [SD] change in BL: −18.24 [14.62] dB; 42 eyes). A slightly lower improvement was observed in years 1 and 2. However, the data available for these timepoints were limited (Figure 1). A numerically higher improvement in light sensitivity was observed in patients aged <18 years compared to those aged ≥18 years at all time points with a mean (SD) change from BLs of −29.80 (1.27) dB (2 eyes) dB vs. −12.35 (13.64) dB (8 eyes) and −17.12 (18.44) dB (3 eyes) vs. −12.63 (24.52); dB (10 eyes) at years 1 and 2, respectively (Figure 1). However, the data were only available for a very limited number of patients for this comparison.

The mean BL VA (LogMAR) was 1.14 (0.57; 148 eyes). No clinically meaningful change in the mean VA was observed from BL up to year 2 with data from fewer patients available post-VN treatment (mean [SD] change from BL: −0.03 [0.55; 24 eyes]; Figure 2). Patients aged <18 years had some improvement (mean [SD] change in BL of 0.15 [0.27] at month 1 (45 eyes), −0.20 [0.30] at month 6 (29 eyes), and −0.16 [0.13] at year 1 (7 eyes) in VA post-VN administration when compared to adult patients (mean (SD) change in BL of −0.01 [0.29] at month 1 (44 eyes), 0.02 (0.30) at month 6 (30 eyes), and −0.02 [0.18] at year 1 (12 eyes)) at all timepoints except year 2 (<18-years-old group: −0.02 [0.69], 12 eyes; ≥18-years-old group: −0.04 [0.40], 12 eyes) (Figure 2).

The mean (SD) foveal thickness on the OCT at BL was 209.2 (45.82) µm (117 eyes). No clinically significant change in BL was observed in the mean (SD) foveal thickness up to year 2 (−22.20 [41.92] µm; 19 eyes; Figure 3) with an inconsistency of the data availability at all time points (Figure 3).

About 98.6% (136 of 138) and 97.2% (70 of 72) of the eyes showed no subretinal fluid (SRF) on the OCT at BL and at month 1, respectively. SRF was absent from all eyes at all later time points (month 6 [58 eyes], year 1 [20 eyes], and year 2 [19 eyes]). Intraretinal fluid was absent during all timepoints, except in 1 eye at year 2. The outer nuclear layer was present in 80.7% (109 of 135) of the eyes at BL, 79.7% (59 of 74) at month 1, 81.8% (45 of 55) at month 6, 57.9% (11 of 19) at year 1, and 73.7% (14 of 19) at year 2.

The baseline data for VF (Octopus kinetic perimetry (V4e)) were available only for 59 out of 183 treated eyes. Overall, the change in BL in the VF remained positive up to year 2 (Appendix A). The median (range) percentage changes in BL in the VF (square degrees) at month 1, month 6, year 1, and year 2 were 7.01 (−86.8 to 727.1; 44 eyes), 24.25 (−84.0 to 284.4; 23 eyes), −3.5 (–63.4 to 155.7; 8 eyes), and 24.66 (−49.1 to 351.4; 11 eyes), respectively (Appendix A). The VF data were highly variable because of the inconsistency of the data availability at different time points.

### 3.4. Safety

Thirty-five patients (34%; 50 eyes) experienced ocular TEAEs (Table 2). The most common ocular TEAEs were related to CRA (at the injection site and/or elsewhere) reported in 13 patients (12.6%; 19 eyes [10.4%]) post-treatment. The CRA was used in this study as a grouping of the following TEAEs: retinal degeneration, retinal depigmentation, and atrophy at the injection site. Nine patients with CRAs received VN bilaterally and six of them had CRAs in both eyes. Four of the patients with CRAs received VN unilaterally. Twelve patients (18 eyes) developed atrophy at the injection site, whereas five patients (8 eyes) had retinal degeneration reported as “progressive atrophic changes in the retina” (four of five patients also had injection-site atrophy). In three patients, the TEAEs of both injection-site atrophy and retinal degeneration were reported in both eyes. All CRA events reported were of mild severity.

Eighteen patients (17.5%) experienced ocular TEAESIs (24 eyes). The most frequently reported terms of interest were “intraocular inflammation and/or infection related to procedure” (n = 7 [6.8%]; 10 eyes [5.5%]), followed by “increased intraocular pressure” (IOP; n = 5 [4.9%]; 7 eyes [3.8%]) and “foveal thinning” (n = 4 (3.9%); 5 eyes (2.7%)) (Table 3). Of note, n = 4 patients (6 eyes) had increased IOPs related to corticosteroids. A TEAESI “retinal tear” was reported in two patients (two eyes), and these events were successfully managed with cryotherapy. Eight patients experienced non-ocular TEAEs (14 events); the most frequent was a headache (n = 4; 6 events). The other TEAEs reported were psychiatric disorders (n = 1; 3 events), cardiovascular disorders, vertigo, falls, injuries (n = 1, 1 event, each), and the complication of pregnancy (1 event (headache)).

Two patients experienced ocular serious TEAEs (three events): one patient experienced eye inflammation in both eyes (moderate; suspected to be related to VN and the sub-retinal administration procedure), while the other patient had an increased IOP of moderate severity in one eye (moderate; suspected to be related to VN, the sub-retinal administration procedure, and peri-operative steroids). Non-ocular serious TEAEs occurred in one patient who had no history of a psychiatric disorder: aggression, agitation, or delirium (one event of each). These events were suspected to be related to perioperative steroid treatment. No other non-ocular serious TEAEs were reported.

### 3.5. Results of Patients with CRAs

The mean age of patients with CRAs (*n* = 13; 19 eyes) was 20.5 (range: 9–33) years and 53.8% were males. Most patients affected were adults (*n* = 8; 61.5%). All patients with CRAs received the recommended dose of 300 µL of VN. A higher proportion of eyes with CRAs were treated using an automated injection system (10 of 19; 52.6%) than eyes without CRAs (33 of 164; 20.1%). The use of an automated injection system was reported for two sites. The mean (SD) time between the treatment of the two eyes was longer (161.9 [149.09] days) in eyes with CRAs than in eyes without CRAs (21.3 [54.44] days). The data indicated no relationship between the occurrence of CRAs and the first/second eye treatments. Time to onset (mean [range]) for injection-site atrophy was 27.5 (1–695) days, whereas that for progressive atrophic changes (retinal degeneration) was 101.5 (34–212) days.

A clinically meaningful increase in the mean FST (white light) was observed in patients with and without CRAs post BL. In patients with CRAs, the mean (SD) change in BL was −9.90 (6.28) dB at month 1 (8 eyes), −13.32 (9.16) dB at month 6 (11 eyes), −9.45 (8.07) dB at year 1 (5 eyes), and −18.00 dB at year 2 (1 eye). However, a greater increase in white-light sensitivity was observed in patients without CRAs than those with CRAs at all timepoints, except year 2; the mean (SD) change in FST (white light) from BL was −17.83 (14.13) dB at month 1 (43 eyes), −19.98 (15.87) dB at month 6 (31 eyes), −22.23 (16.74) dB at year 1 (5 eyes), and −13.30 (23.58) dB at year 2 (12 eyes) (Figure 4). The mean VA at BL was 0.89 (0.50) LogMAR in 19 eyes with CRAs compared to 1.18 (0.57) LogMAR in 129 eyes without CRAs. Overall, no meaningful change in BL was observed up to year 2 (Figure 5).

### 3.6. Pregnancy

Pregnancy was reported for three patients (two females, one partner of a male patient). Pregnancy follow-up data were not available as consent was not given to allow for the collection of such information.

## 4. Discussion

PERCEIVE is the first ex-US and largest registry-based study to date, evaluating the real-world safety and effectiveness of VN in patients with biallelic *RPE65* mutation-associated IRDs. There is another ongoing post-authorization, registry-based study assessing the long-term safety of VN in patients with IRDs in the United States (NCT03597399) [22].

Overall, the safety and effectiveness of VN, observed in the PERCEIVE study up to 2 years, including sustained FST improvement, are consistent with the findings of pivotal VN clinical trials [13,14,15]. The increase in white-light sensitivity at month 6 of −18.24 dB in PERCEIVE, corresponding to −1.82 log_10_(cd.s/m^2^), was comparable to the results of the pivotal phase 3 trial at month 6 (−2.45 [1.48] log_10_[cd.s/m^2^]) [23]. There was no meaningful change in the BCVA and mean foveal thickness reported up to year 2. The age range of patients in the PERCEIVE study (2–51 years) was comparable to that of the patients in pivotal VN trials (4–46 years); however, PERCEIVE included four patients below the age of 4 years who had similar safety observations as those reported for the overall population [15]. Moreover, the baseline mean (range) BCVA in PERCEIVE was 1.14 (0.20–2.60) LogMAR compared to 1.08 (0.52–2.06) LogMAR reported in the phase 3 trial [23].

These results are also in concordance with the recent real-world study by Deng et al. who evaluated 27 eyes of VN-treated pediatric patients with IRDs, and observed significant improvements in the FST, VA, and VF, maintained up to 1-year post-treatment [20]. In addition, similar improvements in the FST after VN administration were observed in the LIGHT study [16]. In a case series, significant improvements in FSTs were also reported, and a trend toward the improvement of the BCVA was observed in pediatric patients after treatment with VN, although no statistically significant change in the BCVA was reported for adults [24]. In this study, although the data were available only from a limited number of patients, a greater improvement in white-light sensitivity was observed in patients aged <18 years than in adult patients (Figure 1). This could be due to the presence of a higher number of viable retinal cells in pediatric and adolescent patients compared to adult patients and highlights the importance of the early diagnosis and treatment of patients with IRD. While age group-wise comparisons for mean foveal thickness showed no consistent trend (Figure 3), we were cautious when interpreting a single-point measurement of foveal thickness as it may have not reflected the health of the retina outside the limits of the single foveal B scan on the OCT.

Given that PERCEIVE is an observational, real-world study, the VF data collected are variable and difficult to interpret due to the heterogeneity of the devices used and the lack of standardized data acquisition and interpretation protocols. Additionally, VF data were available only for a few patients at BL. This could show that the VF assessment was not performed as the standard of care in IRD patients at most sites. The other plausible reasons for the variability could be the complexity of the examination, attention of the patient, and the operator’s experience.

Although pregnancies were reported at the centers, consent was not given for the collection of pregnancy-related details.

Chorioretinal atrophy was identified as a new adverse drug reaction, which was not described in pivotal VN trials as an adverse event (AE). The potential factors that could have contributed to patients developing RPE changes and CRA included the natural progression of the disease, potential toxicity of the AAV2 vector in retinal cells, immunogenicity of the vector, surgical administration procedures, and/or ocular factors, such as the presence of myopia [17,18]. In general, CRA was observed at the following three locations: at the injection site, in the treatment area (bleb and confluent, immediate vicinity), and in the retinal periphery (away from the treatment area) [18]. While the etiology remains to be identified and confirmed, these observations may represent manifestations of very different mechanisms, or a consequence of contributing factors, such as mechanical stress in the RPE and photoreceptor cells (shearing off), the natural progression of IRD, and others [18,25]. Notably, the ellipsoid zone was absent from 18 treated eyes (14.1%) at BL, which could have been due to the presence of outer retinal atrophy [26]. However, the loss of visual function associated with the CRA event was not reported at the time of the data cut-off. The findings observed in PERCEIVE are in line with other published reports of development of retinal atrophy after VN administration [17,18]. In both these reports, CRA or retinal atrophy did not appear to affect VN therapy-induced improvements in vision outcomes (BCVA, FST, and VF) [18,19]. Nevertheless, there is a potential risk that further progression with the enlargement of atrophic lesions can cause additional functional impairments in such patients.

The post-authorization long-term follow-up treatment of patients, who received gene therapies, is required to collect the data on any delayed AEs [27]. PERCEIVE is the largest real-world gene therapy study to date, with a long-term follow-up outcome of treated patients (mandated by the health authorities). The data from this observational study portray a diverse, multinational population under real-world conditions. Despite the impact of the COVID-19 pandemic on the frequency of follow-up visits, sites were able to continue treating and enrolling patients in the study. Long-term follow-up visits will allow the collection of further information on the safety and clinical effect of VN.

This study ad some limitations. Although there was a standard treatment protocol per prescribing information, the actual treatment procedure could varied among sites. Moreover, as this was an observational study, the lack of a dedicated clinical trial protocol, standard operating procedures, and a central reading center; differences in the standard of care; and variable resources led to differences in the outcomes of follow-up visits and data collected at various study sites. The absence of a central reading center could have led to the varied reporting and interpretation of data, including morphological features, such as CRA. Although real-world studies have better generalizability compared to clinical trial data, a synchronization of the follow-up guidelines or procedures to improve the evaluation of retinal images and data collection is required to achieve better insights from such observational studies.

This non-interventional post-approval study was a key contribution to the field of IRD treatments as it was the world’s first and largest systematic real-world data collection of patients treated with the world’s first approved ocular gene therapy. Furthermore, it not only confirmed the efficacy of treatment from the phase I–III studies in clinical practice, but, significantly, highlighted the benefits of a post-approval registry to help identify an important safety signal that was of great interest to ophthalmologists and, more broadly, to other ongoing ocular gene therapy studies. This is even more important as ocular gene therapy is used to target chronic and prevalent diseases, such as AMD.

## 5. Conclusions

PERCEIVE is the largest study to date of patients with biallelic *RPE65*-associated IRDs who were treated with ocular gene therapy. Overall, the up to 2-year results from PERCEIVE demonstrate the safety and effectiveness of VN in real-word settings, which is consistent with the findings from the VN clinical trials. VN therapy led to sustained improvements in the FSTs for pediatric and adult patients, which was observed to be better in patients aged <18 years than adult patients, highlighting the importance of the early treatment of patients with IRDs. Chorioretinal atrophy was ben identified as a new adverse drug reaction from post-marketing reports, including events reported within this study. Notably, CRA events did not impact the benefits of VN treatment till the data cut-off. These events will continue to be monitored and more accurately defined to determine any long-term impact on visual function in these patients and help identify potential causes, so that efforts can be made to mitigate these undesirable effects in the future. As the study is ongoing, PERCEIVE will continue to collect valuable information and provide real-world evidence on the long-term safety and effectiveness of VN in the real-world scenario.

## Figures and Tables

**Figure 1 biomolecules-14-00122-f001:**
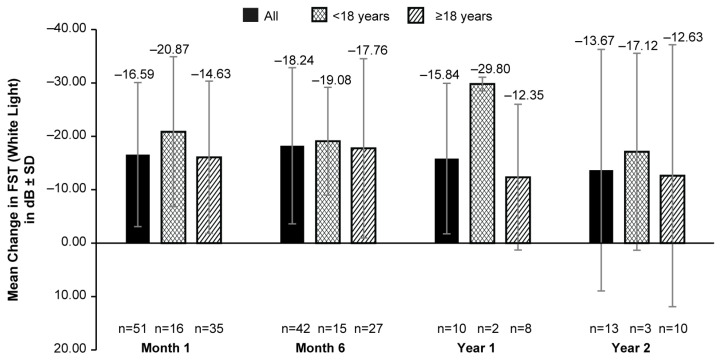
Mean change from baseline in FST (white light) up to 2 years after VN administration in all patients, patients aged <18 years and those aged ≥18 years. dB, decibel; FST, full-field light-sensitivity threshold testing; n, number of eyes; VN, voretigene neparvovec. Baseline data were available for 127 eyes. Error bars represent SD.

**Figure 2 biomolecules-14-00122-f002:**
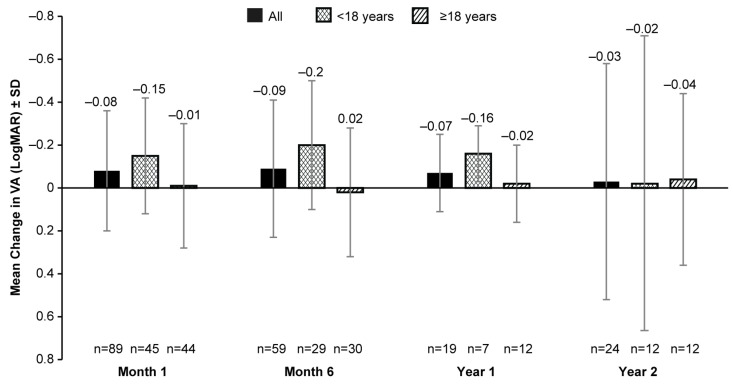
Mean change from baseline in VA (LogMAR) up to 2 years after receiving VN in all patients, patients aged <18 years, and those aged ≥18 years. VA, visual acuity; LogMAR, logarithm of the minimum angle of resolution; n, number of eyes; VN, voretigene neparvovec. Baseline data were available for 148 eyes. Error bars represent SD.

**Figure 3 biomolecules-14-00122-f003:**
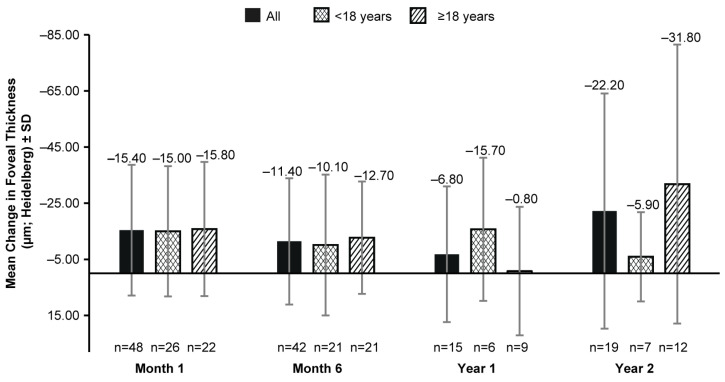
Mean change from baseline in foveal thickness (µm; Heidelberg Spectralis OCT) up to 2 years after VN therapy in all patients, patients aged <18 years, and those aged ≥18 years. n, number of eyes; VN, voretigene neparvovec. Baseline data were available for 117 eyes. Error bars represent SD.

**Figure 4 biomolecules-14-00122-f004:**
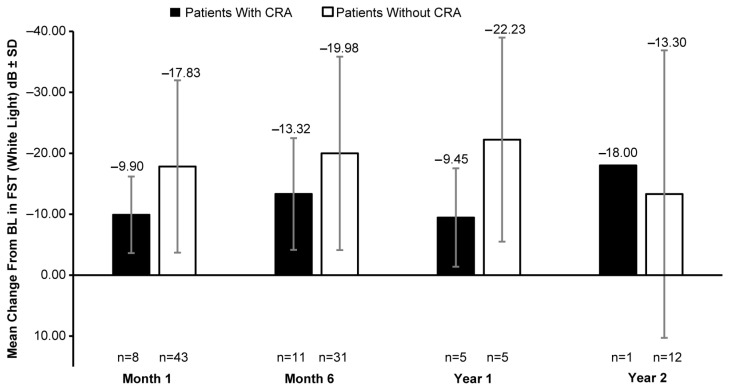
Mean change in FST (white light) from baseline in patients with CRAs versus those without CRAs. BL, baseline; dB, decibel; CRA, chorioretinal atrophy; FST, full-field light-sensitivity threshold testing; n, number of eyes. Baseline data are available for 15 eyes with CRAs and 112 eyes without CRAs. Error bars indicate SD.

**Figure 5 biomolecules-14-00122-f005:**
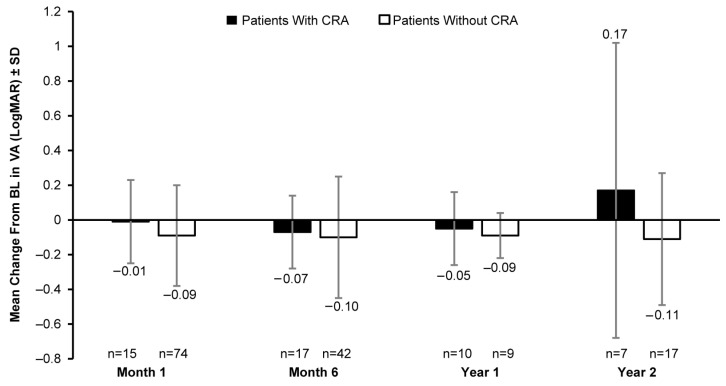
Mean change in VA (LogMAR) from baseline in patients with CRAs versus those without CRAs. VA, visual acuity; BL, baseline; CRA, chorioretinal atrophy; LogMAR, logarithm of minimum angle of resolution; n, number of eyes. Baseline data are available for 19 eyes with CRAs and 129 eyes without CRAs. Error bars indicate SD.

**Table 1 biomolecules-14-00122-t001:** Demographics and baseline characteristics of the study population.

Variables	FAS, *N* = 103
**Mean (SD) age *, years**	19.5 (10.85)
**Age * group, n (%)**	
2–11 years	31 (30.1)
12–17 years	20 (19.4)
≥18 years	52 (50.5)
**Sex, n (%)**	
Males	50 (48.5)
Female	52 (50.5)
Unknown	1 (1.0)
**Race, n (%)**	
White	72 (69.9)
Black or African American	4 (3.9)
Asian	3 (2.9)
Indian	2 (1.9)
Unknown	24 (23.3)
**Ethnicity, n (%)**	
Hispanic/Latino	2 (1.9)
Not Hispanic/Latino	74 (71.8)
Not reported	22 (21.4)
Unknown	5 (4.9)
**Country, n (%)**	
Germany	24 (23.3)
France	21 (20.4)
Italy	15 (14.6)
United Kingdom	11 (10.7)
Denmark	10 (9.7)
Croatia	4 (3.9)
Israel	4 (3.9)
Spain	3 (2.9)
Austria	2 (1.9)
Belgium	2 (1.9)
Czech Republic	2 (1.9)
Slovakia	2 (1.9)
Brazil	1 (1)
Netherlands	1 (1)
Poland	1 (1)

* Age at the time of VN administration. Percentages are calculated based on n. FAS, full analysis set; N, total number of patients in FAS; n, number of patients in given category; SD, standard deviation.

**Table 2 biomolecules-14-00122-t002:** Ocular treatment-emergent AEs by primary SOC and preferred term.

Primary SOC/Preferred Term	Patients, *N* = 103 n (%)	Treated Eyes, *N* = 183 n (%)	Number of Events
**At least one event**	35 (34.0)	50 (27.3)	75
**Eye disorders**	27 (26.2)	35 (19.1)	45
Metamorphopsia	5 (4.9)	5 (2.7)	5
Retinal degeneration	5 (4.9)	8 (4.4)	8
Foveal degeneration	4 (3.9)	5 (2.7)	5
Vitritis	4 (3.9)	6 (3.3)	6
Eye inflammation	3 (2.9)	4 (2.2)	4
Retinal tear	2 (1.9)	2 (1.1)	2
Vitreous hemorrhage	2 (1.9)	2 (1.1)	2
Cataract	1 (1.0)	1 (0.5)	1
Dyschromatopsia	1 (1.0)	1 (0.5)	1
Eye pain	1 (1.0)	1 (0.5)	1
Glare	1 (1.0)	1 (0.5)	1
Macular degeneration	1 (1.0)	1 (0.5)	1
Macular fibrosis	1 (1.0)	1 (0.5)	1
Macular hole	1 (1.0)	1 (0.5)	1
Macular scar	1 (1.0)	1 (0.5)	1
Photophobia	1 (1.0)	1 (0.5)	1
Retinoschisis	1 (1.0)	1 (0.5)	1
Visual acuity reduced	1 (1.0)	2 (1.1)	2
Visual field defect	1 (1.0)	1 (0.5)	1
**General disorders and administration-site conditions**	12 (11.7)	18 (9.8)	18
Injection-site atrophy	12 (11.7)	18 (9.8)	18
**Investigations**	5 (4.9)	7 (3.8)	7
Increased intraocular pressure	5 (4.9)	7 (3.8)	7
**Injury, poisoning, and procedural complications**	4 (3.9)	5 (2.7)	5
Intentional product-use issue	1 (1.0)	2 (1.1)	2
Post-procedural complication	1 (1.0)	1 (0.5)	1
Procedural pain	1 (1.0)	1 (0.5)	1
Suture-related complication	1 (1.0)	1 (0.5)	1

Full analysis set. A patient/treated eye with multiple occurrences of an AE for a system organ class (SOC) or preferred term is counted only once for each SOC/preferred term in column n. Each eye treated with voretigene neparvovec is included in the treated-eyes columns. Preferred terms are sorted within SOCs by descending frequency in the “Patients” column. MedDRA Version 24.0 was used for the reporting of AEs. AE, adverse event; MedDRA, Medical Dictionary for Regulatory Activities; N, total number; n, number under the given category; SOC, system organ class.

**Table 3 biomolecules-14-00122-t003:** Ocular treatment-emergent AEs of special interest by term of interest and preferred term.

Term of Interest/Preferred Term	Patients, *N* = 103 n (%)	Treated Eyes, *N* = 183 n (%)	Number of Events
**Any ocular AEs of special interest**	18 (17.5)	24 (13.1)	29
**Intraocular inflammation and or infection related to procedure**	7 (6.8)	10 (5.5)	10
Vitritis	4 (3.9)	6 (3.3)	6
Eye inflammation	3 (2.9)	4 (2.2)	4
**Increased intraocular pressure**	5 (4.9)	7 (3.8)	7
Increased intraocular pressure	5 (4.9)	7 (3.8)	7
**Foveal thinning**	4 (3.9)	5 (2.7)	6
Foveal degeneration	4 (3.9)	5 (2.7)	5
Macular scar	1 (1.0)	1 (0.5)	1
**Retinal tear**	2 (1.9)	2 (1.1)	2
Retinal tear	2 (1.9)	2 (1.1)	2
**Maculopathy (e.g., epiretinal membrane, and macular pucker)**	2 (1.9)	2 (1.1)	2
Macular fibrosis	1 (1.0)	1 (0.5)	1
Retinoschisis	1 (1.0)	1 (0.5)	1
**Macular hole**	1 (1.0)	1 (0.5)	1
Macular hole	1 (1.0)	1 (0.5)	1
**Cataract**	1 (1.0)	1 (0.5)	1
Cataract	1 (1.0)	1 (0.5)	1

Full analysis set. A patient/treated eye with multiple occurrences of a term of interest is counted only once for that term of interest in column n. Each eye treated with voretigene neparvovec is included in the treated-eyes columns. Terms of interest are presented in alphabetical order. Adverse events of special interest are collected through a standardized safety questionnaire. MedDRA Version 24.0 is used for the reporting of AEs. AE, adverse event; MedDRA, Medical Dictionary for Regulatory Activities; N, total number; n, number under the given category.

## Data Availability

Data are available upon reasonable request. All data relevant to the study are included in the article or uploaded as Appendix A. We confirm that the data generated by our research support our current article. Novartis is committed to sharing with qualified external researchers, providing access to patient-level data, and supporting clinical documents from eligible studies. These requests were reviewed and approved by an independent review panel based on scientific merit. All data provided are anonymized to respect the privacy of patients. The authors confirm they had no special access or privileges that other researchers would not have.

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
