# Peer review of "Real-World Safety and Effectiveness of Voretigene Neparvovec: Results up to 2 Years from the Prospective, Registry-Based PERCEIVE Study"

_biomolecules, 2024, doi:10.3390/biom14010122_

Round 1

Reviewer 1 Report

Comments and Suggestions for Authors

Overall, this is a very well written paper with a few minor comments. 

In the results section, the authors mention that The outer nuclear layer was present in 80.7% (109 of 135) of the eyes at BL, 79.7% (59 of 74) at month 1, 81.8% (45 of 55) at month 6, 57.9% (11 of 19) at year 1, and 73.7% (14 of 19) at year 2. I understand that there are some inconsistent findings across different time points due to small sample size but can the authors speculate any other reasons why there was smaller percentage of patients with the presence of outer nuclear layer in year 1 and again higher percentage in year 2. Furthermore, it would be great if the authors can explain the meaning behind the presence of the outer nuclear layer - does it mean the rod and cones are surviving ? This could be explained in the discussion.

Again, the results of the visual field findings are inconsistent across different time points particularly the year 1 results. I assume this could also be due to small sample size and the large variability in the data. It could be beneficial to present the data categorically in groups - % of patients who had positive change in VF sensitivity, % with no change and % with negative change. This way we can see how many patients actually benefitted from the treatment. 

Again, the authors mention there was no meaningful change in BCVA up to 2 years but can the authors categorize what percentage of patients had actual improvement in visual acuity (if any and if data is available) and discuss associated findings among those whose vision improved vs whose deteriorated or remained stable. 

The authors also mention that there was no significant impact of CRA on visual function that could be because ? the CRA lesions were not within the foveal region, do the authors have any data on microperimetry or multifocal ERG  to assess the level of retinal sensitivity in these areas. If not, are there any plans to include such measures.

The authors mention in the methods section that they collected data on the patient/caregiver reported outcomes (mVFQ-25 and BIC) but I did not see the results. Some discussion on why this was not included would be appreciated. 

Author Response

Overall, this is a very well written paper with a few minor comments. 

Comment 1:

In the results section, the authors mention that The outer nuclear layer was present in 80.7% (109 of 135) of the eyes at BL, 79.7% (59 of 74) at month 1, 81.8% (45 of 55) at month 6, 57.9% (11 of 19) at year 1, and 73.7% (14 of 19) at year 2. I understand that there are some inconsistent findings across different time points due to small sample size but can the authors speculate any other reasons why there was smaller percentage of patients with the presence of outer nuclear layer in year 1 and again higher percentage in year 2. Furthermore, it would be great if the authors can explain the meaning behind the presence of the outer nuclear layer - does it mean the rod and cones are surviving? This could be explained in the discussion.

Author’s Response:

PERCEIVE is an observational study and as such has no pre-defined or mandated visit schedule and data were collected from assessments performed as part of medical practice. While such studies produce highly informative and relevant real-world evidence, the data acquisition does not follow standardized protocols. We share your observation that there are some inconsistent findings which may largely be due to the small sample size together with the respective data not being available for all time points and differences in the investigator-reported assessment of the presence/absence of the outer nuclear layer.

Change to the manuscript:

Page 2, line 86-87 We have added the same in the introduction “with no pre-defined or mandated visit schedule,” for clarity. Based on these aspects we want to refrain from speculative interpretations regarding the outer nuclear layer.

Comment 2:

Again, the results of the visual field findings are inconsistent across different time points particularly the year 1 results. I assume this could also be due to small sample size and the large variability in the data. It could be beneficial to present the data categorically in groups - % of patients who had positive change in VF sensitivity, % with no change and % with negative change. This way we can see how many patients actually benefitted from the treatment. 

Author’s Response:

We share your observation that the reported visual field data are variable, reflective of limitations such as the small sample size and other factors, such as lack of standardized acquisition and interpretation protocols, non-comparable devices etc., as mentioned above. With respect to the visual field we suspect that progression and state of the disease at the time of treatment may also contribute to this. It is also noteworthy, that you would expect the retinal dystrophy to progress outside of the treated area as part of natural disease progression, resulting in additional complexity when trying to assess this data. While an increase in visual field can be observed in patients treated with voretigene neparvovec, the main benefit is an increase in light sensitivity within the treated retinal area, which may be associated with or without the benefit of an increased visual field. Given the quality of visual field data and limited numbers, we would not be able to derive meaningful conclusions from this data. We therefore want to keep the reporting of the data descriptive.

Change to the manuscript:

No change made.

Comment 3:

Again, the authors mention there was no meaningful change in BCVA up to 2 years but can the authors categorize what percentage of patients had actual improvement in visual acuity (if any and if data is available) and discuss associated findings among those whose vision improved vs whose deteriorated or remained stable. 

 Author’s Response:

A categorical analysis was not performed and therefore no correlation to other findings is available. Data from the pivotal phase 3 study indicates that there is a good relationship between the observed MLMT and FST improvements, while there is no good correlation with visual acuity. This possibly reflects voretigene neparvovec’s rod-mediated mechanism of action, while visual acuity is primarily cone-mediated.

Change to the manuscript:

No change made.

Comment 4:

The authors also mention that there was no significant impact of CRA on visual function that could be because? the CRA lesions were not within the foveal region, do the authors have any data on microperimetry or multifocal ERG to assess the level of retinal sensitivity in these areas. If not, are there any plans to include such measures.

 Author’s Response:

The PERCEIVE study is only collecting data from assessments performed as part of standard medical practice and as an observational study cannot mandate any assessments to be performed or defined/standardized protocols to be followed when conducting assessments. While the study allows for the collection of limited microperimetry data, there is no collection of images or central analysis of imaging data in the absence of standardized imaging protocols, which would allow to determine the exact location of any reported CRA and analysis in connection with, e.g. microperimetry data. The currently collected microperimetry data is also insufficient for such analyses to be performed. Given the limitations of a non-interventional PASS study, there are no plans to collect/analyze such data at this point.

Change to the manuscript:

No change made.

Comment 5:

The authors mention in the methods section that they collected data on the patient/caregiver reported outcomes (mVFQ-25 and BIC) but I did not see the results. Some discussion on why this was not included would be appreciated. 

Author’s Response:

While data for mVFQ-25 and BIC are being collected in the study, there was insufficient data available at the time for analysis. This is largely attributed to the fact that patients initially enrolled into the PERCEIVE study were largely retrospective patients, i.e. patients who had at least one eye treated before joining the study. Such patients are lacking baseline data and do not meet the minimum requirements for analysis. Adoption of the questionnaires has also been mixed, as the study does not mandate them to be performed and they are not part of standard medical practice. Taken together, this contributed for this data to be yet too limited at this stage to be analyzed and reported. However, this data will be reported when more data has become available during the conduct of the study. We have revised the manuscript to not refer to the collection of mVFQ-25 and BIC, as the collected data was too limited for an analysis to be performed. Thank you for your comment and observation.

Change to the manuscript:

Page 3, line 124-127; Page 3, lines 134-135; page 4, line 150-151; Page 11, lines 342-344

We have deleted the sentence “patient/caregiver reported outcomes (mVFQ-25 and BIC) questionnaires,

Reviewer 2 Report

Comments and Suggestions for Authors

This reads well and I only have very minor comments. It is clearly important to publish these results, which are well-presented.

Abstract

This states changes in BCVA are not significant. It would be worthwhile stating the changes, if space allows.

Introduction

The prevalence estimate (1 in 300,000) is probably at the upper end of the real estimate: it might be worthwhile stating an imputed range.

Treatment intervention

It’s unclear what is meant by “limiting dissection”? Is this the creation of a break in the ILM?

Effectiveness

The authors state that the lower improvements in FST at later timepoints reflects limited data. It’s unclear what the basis of this assertion is: it would be prudent to present something which reflects this (there are of course alternative possibilities, e.g. waning of efficacy etc etc).

Discussion

The text states that a change in FST of -18.24dB change in FST is equivalent to -2.82log10 cd.s/sq.m. This doesn’t make sense. Surely it should be a -1.82log10 cd.s/sq.m change.

Author Response

This reads well and I only have very minor comments. It is clearly important to publish these results, which are well-presented.

Comment 1:

Abstract

This states changes in BCVA are not significant. It would be worthwhile stating the changes, if space allows.

Author’s Response: Thank you for your suggestions, however we are not able to add the BCVA changes due to word count limit.

Comment 2:

Introduction

The prevalence estimate (1 in 300,000) is probably at the upper end of the real estimate: it might be worthwhile stating an imputed range.

Author’s Response: We have added the range as suggested.

Change to the manuscript:

Page 2, line 53. ‘1.20-2.37 per 100,000’ added. With new reference being added and other references are updated accordingly throughout the manuscript.

Comment 3:

Treatment intervention

 It’s unclear what is meant by “limiting dissection”? Is this the creation of a break in the ILM?

Author’s Response:

The study is collecting deviations from the standard procedure (per prescribing information) in a standardized manner, however other deviations may be reported as verbatim (non-categorized text). While we have reported the verbatim text, we believe that “limiting dissection” may refer to limiting membrane peeling performed at the retinotomy site. However, there are no further details available. As we are reporting here the verbatim text of reporting descriptively, we do not want to give more insights on the meaning.

Change to the manuscript:

No change made.

Comment 4:

Effectiveness

The authors state that the lower improvements in FST at later timepoints reflects limited data. It’s unclear what the basis of this assertion is: it would be prudent to present something which reflects this (there are of course alternative possibilities, e.g. waning of efficacy etc etc).

Author’s Response:

We agree that a lower improvement in FST at later timepoints may reflect waning of efficacy. However, the limited data availability does not allow for this conclusion to be made. The long-term follow-up data from the pivotal phase 3 study indicates that efficacy is maintained for the majority of patients. In our opinion, the limited long-term data available from the PASS study does not allow for final conclusions to be made about the duration of benefit for eyes treated with voretigene neparvovec.

Change to the manuscript:

Page 6, line 215-216: We have revised our statement in the manuscript with respect to the limited data availability.

Comment 5:

 Discussion

The text states that a change in FST of -18.24dB change in FST is equivalent to -2.82log10 cd.s/sq.m. This doesn’t make sense. Surely it should be a -1.82log10 cd.s/sq.m change.

Author’s Response:

Thank you for making us aware of this error, which was possibly a typo, as the data was originally reported in dB and only converted for comparison reasons. We have changed it to -1.82log10 cd.s/sq.

Change to the manuscript:

Page 12, line 353: We have changed it to -1.82log10 cd.s/sq.

Reviewer 3 Report

Comments and Suggestions for Authors

This study enrolled a large cohort of RPE65 associated cases with VN treatment with rich data. I would like to know more about the BCVA chart. 

Author Response

Comment 1:

This study enrolled a large cohort of RPE65 associated cases with VN treatment with rich data. I would like to know more about the BCVA chart.

Author’s Response:

Visual acuity was assessed as per standard of care and converted to logarithm of the minimum angle of resolution (LogMAR) scores for analysis. Off-chart vision measurements were assigned LogMAR values using the scale adapted from Lange 2009. As an observational study, the study did not mandate the visual acuity assessments to be performed in a standardized way, e.g. by using ETDRS charts in connection with standardized lighting conditions or specifically trained assessors, as it is common for highly standardized interventional studies. In addition, the young age of some of the patients, as well as the inclusion of older patients with advanced disease, did not allow for visual acuity assessments in some patients, or require off-chart assessments, like counting fingers, hand movements, etc. in order to provide estimates for visual acuity. The term BCVA has been replaced by visual acuity to avoid confusion.

Change to the manuscript:

This has been corrected throughout the manuscript including the figure no. 2 and figure 5.